# Disruption of Electroencephalogram Coherence between Cortex/Striatum and Midbrain Dopaminergic Regions in the Knock-Out Mice with Combined Loss of Alpha, Beta, and Gamma Synucleins

**DOI:** 10.3390/biomedicines12040881

**Published:** 2024-04-16

**Authors:** Vasily Vorobyov, Alexander Deev, Kirill Chaprov, Natalia Ninkina

**Affiliations:** 1Institute of Cell Biophysics, Russian Academy of Sciences, 142290 Pushchino, Russia; vorobyovv2@gmail.com; 2Institute of Theoretical and Experimental Biophysics, Russian Academy of Sciences, 142290 Pushchino, Russia; aadeev@gmail.com; 3Institute of Physiologically Active Compounds at Federal Research Center of Problems of Chemical Physics and Medicinal Chemistry, Russian Academy of Sciences, 142432 Chernogolovka, Russia; chaprov@ipac.ac.ru; 4School of Biosciences, Cardiff University, Sir Martin Evans Building, Museum Avenue, Cardiff CF10 3AX, UK

**Keywords:** EEG, functional connectivity, forebrain, midbrain, knock-out, mouse

## Abstract

The malfunctioning of the brain synucleins is associated with pathogenesis of Parkinson’s disease. Synucleins’ ability to modulate various pre-synaptic processes suggests their modifying effects on the electroencephalogram (EEG) recorded from different brain structures. Disturbances in interrelations between them are critical for the onset and evolution of neurodegenerative diseases. Recently, we have shown that, in mice lacking several synucleins, differences between the frequency spectra of EEG from different brain structures are correlated with specificity of synucleins’ combinations. Given that EEG spectra are indirect characteristics of inter-structural relations, in this study, we analyzed a coherence of instantaneous values for EEGs recorded from different structures as a direct measure of “functional connectivity” between them. Methods: EEG data from seven groups of knock-out (KO) mice with combined deletions of alpha, beta, and gamma synucleins versus a group of wild-type (WT) mice were compared. EEG coherence was estimated between the cortex (MC), putamen (Pt), ventral tegmental area (VTA), and substantia nigra (SN) in all combinations. Results: EEG coherence suppression, predominantly in the beta frequency band, was observed in KO mice versus WT littermates. The suppression was minimal in MC-Pt and VTA-SN interrelations in all KO groups and in all inter-structural relations in mice lacking either all synucleins or only beta synuclein. In other combinations of deleted synucleins, significant EEG coherence suppression in KO mice was dominant in relations with VTA and SN. Conclusion: Deletions of the synucleins produced significant attenuation of intra-cerebral EEG coherence depending on the imbalance of different types of synucleins.

## 1. Introduction

Parkinson’s disease (PD) is considered a type of “alpha-synucleinopaties” [1] and associated with the deposition of aggregated alpha synuclein (alpha-syn) [2]. Alpha-syn is a major member of vertebrate-specific proteins family also including beta- and gamma-syns [3]. Overlapping expression of different types of synucleins and their location in presynaptic terminals [4] suggest a redundancy of synaptic functions of the synucleins that in turn highlights a compensatory potential of this composition of synucleins. Indeed, increased expression of the remaining family member (beta-syn) has been shown to initiate compensatory/protective processes in alpha/gamma-synuclein double-knock-out (KO) mice [5,6]. However, little is known about either the brain function mechanisms affected by the knocking out or the compensatory abilities of different types of synucleins involved in the recovery processes.

Endogenous alpha-syn expression has been shown to be involved in selective regional brain vulnerability [7], which in turn is associated with variations in PD brain connectivity [8,9], thus affecting the intrinsic nets [10]. This can accompanied by disturbed phase relations between rhythmic activities generated in the neuronal circuits with tremendous consequences for the functioning of the brain [11]. Under normal conditions, alpha-syn has been shown to be located predominantly in the presynaptic terminals and on synaptic vesicles membranes [12]. Together, this is supposed to be a basis for the use of the electroencephalogram (EEG) as an effective approach for the monitoring of PD progression and its pharmacological therapy [13]. Thus, EEG recordings from different brain structures allow for the analysis of alpha-syn-associated changes in selective regional brain vulnerability, synaptic integrity, and coordinated network activity affected by neurodegenerative processes [14]. Indeed, in a murine model of amyotrophic lateral sclerosis, we have shown that by measuring the coherence of instantaneous EEG values in different brain structures, we are able to reveal additional details describing “functional connectivity” in the brains of KO mice compared to that in the brains of normal mice (c.f., [15,16]). Recently, EEG frequency analysis in knock-out (KO) mice with various compositions of deleted alpha-, beta-, and gamma-syns has demonstrated an important role of their misbalance in interrelations between EEGs from different brain areas [17]. Generally, this result is expected, given (i) the co-localizations of alpha- and beta-syns in presynaptic terminals in various brain areas [18]; (ii) the gamma-syn-produced modification of interaction of beta-syn with membrane [19]; and (iii) changes in alpha-syn binding to synaptic vesicles under the influence of gamma- and beta-syns [20]. Furthermore, gamma-syn transcription in VTA and SN has been shown to be linked to the release and re-uptake of dopamine (DA) in the nigrostriatal and mesocortical pathways [21]. The necessity of detailed analysis of the mutual influences of the synucleins has been clearly demonstrated in the elimination of MPTP intoxication detrimentally affecting DA system function [6]. This supposedly is associated with the revealed beneficial influence of unfolded (monomeric) clusters of alpha-syns on the mitochondrial function [22]. Thus, a deeper insight into both the intimic functional interrelations between the brain areas, characterized by different levels of coherent activities in various neuronal nets, and coherence modifications in syn-KO mice would be effective in the understanding of the role of synucleins in pathological processes affecting the PD brain. In this study, coherent analysis of the EEG data from our previous work [17] was used to characterize associations between inter-structural “functional connectivity” and different synuclein-dependent types of KO mice. The main aim of this study was to separate the “safe” compositions of deleted syns from “unsafe” ones, i.e., from those producing significant changes in the brains “functional connectivity”. The results obtained from such approach might be a basis for the understanding of mechanisms of syn subtype interactions and their role in the development of PD pathology or/and in its extinction.

The coherence between EEGs recorded from MC, Pt, and DA-producing brain regions (VTA, and SN) was analyzed in adult KO mice with combined deletion of alpha-, beta-, and gamma-syns. KO mice with deleted syns in various combinations were characterized by significant attenuation of intracerebral EEG coherence depending on both the imbalance between different types of syns and inter-structural relations, especially with the dopamine-containing areas, VTA, and SN. Moreover, the lesser vulnerability of VTA versus SN to selective deletion of different types of syns was shown in our study.

## 2. Materials and Methods

### 2.1. Experimental Animals

Details of the section have been considered in our previous paper [17]. Briefly, 84 male mice (3 months old) sorted in KO group by various combinations of deleted alpha (A)-, beta (B)-, and gamma (G)-Syns and wild-type (WT, A+B+G+) littermates were included in this study. Overall, KO mice from seven groups were compared with WT mice (*n* = 10): A-B+G+, A-B-G, A-B+G-, A-B-G+, A+B-G-, A+B-G+, and A+B+G- (*n* = 12, 8, 10, 13, 13, 11, and 7, respectively).

Mice were housed with a 12-h/12-h light/dark cycle, 22–25 °C RT, 50–55% relative humidity, with food and water ad libitum. The procedures were based on the “Guidelines for accommodation and care of animals and the principles of the Directive 2010/63/EU on the protection of animals used for scientific purposes. They were approved by the local Institute Ethics Review Committee (protocol № 48, 15 January 2021) with recommended efforts to minimize the number of the animals and their suffering.

### 2.2. Electrode Implantation and Recording of EEG

After one month’s adaptation, the mice were anesthetized subcutaneously with tiletamine/zolazepam (Zoletil^®^, Virbac, Carros, France, 25 mg/kg) with xylazine solution (Rometar^®^, Bioveta, Ivanovice na Hané, Czech Republic, 2.5 mg/kg). The electrodes for EEG recordings were implanted in MC and Pt (AP: +1.1 mm; ML: ±1.5 mm; DV: −0.75 and −2.75 mm, respectively), in VTA (AP: −3.1, ML: −0.4, DV: −4.5), and in SN (AP: −3.2, ML: +1.3, DV: −4.3) [23]. The electrodes (two insulated 100-µm nichrom wires) were glued together with 100 µm space free from insulation tips. The reference and ground electrodes (0.4 mm stainless steel wire) were placed symmetrically behind the cerebrum. All electrodes were positioned by a computerized 3D stereotaxic StereoDrive (Neurostar, Tübingen, Germany), fixed to the skull, and soldered to a micro-connector. After surgery, mice were contained individually for recovery, followed by the experimental sessions. On day 8, a baseline EEG was recorded at 30 min, starting 20 min after a mouse was placed in the box (see details in [17]).

### 2.3. EEG Spectral Coherence Computation

EEG signals measured between the active and reference electrodes were amplified, filtered (0.1–35 Hz), and sampled (1 kHz) on-line. EEG fragments containing artifacts and epileptic spikes were automatically and manually rejected by the custom-developed software (see [16]). Spectral coherence was estimated by averaging 12-sec epochs of baseline EEGs in the range of 1–30 Hz, with the averaging of data in “classical” EEG bands (in Hz): delta 1 (1–2), delta 2 (2–4), theta (4.0–8.0), alpha (8–12 Hz), beta 1 (12–20), and beta 2 (20–30.0). The values of coherence in each frequency band were averaged at every successive 10-min interval (for further statistical analysis) and totally after 30 min (for illustrations).

### 2.4. Statistics

Differences in the EEG coherency between KO and WT mice were analyzed by two-way ANOVA for repeated measures with Bonferroni’s post hoc test for multiple comparisons (STATISTICA 10; StatSoft, Inc., Tulsa, OK, USA). All sets of data were preliminarily tested on Gaussian distribution and EEG stationarity by use of KPSS test to determine optimal time duration (12 s) of EEG epochs for further coherent analysis. For preliminary evaluation of power and effect size, G*Power version 3.1.9.4 was used (http://www.psycho.uniduesseldorf.de/abteilungen/aap/gpower3, assessed on 6 February 2019). All data are shown as mean ±SEM and considered statistically significant at *p* < 0.05.

## 3. Results

In the baseline period, KO and WT mice behaved similarly, displaying intensive exploration and rare sleep-like bouts. Baseline EEGs in WT (A+B+G+) and KO (A-B+G-) mice (Figure 1) contained patterns of relatively slow EEG in the theta–alpha frequency range, while in the KO mice, the EEG activity in the beta band was more powerfully expressed. Statistical characteristics of baseline EEGs and their frequency compositions in all groups of mice have been described in detail in our previous paper [17].

In mice from the KO and WT groups, the differences between their baseline EEG coherence levels in 10-min intervals (see an example in Figure 2) were relatively stable over a 30 min period (see Figure A1) and characterized by lower levels of EEG coherence in KO mice versus in WT littermates (Figure 3, Figure 4 and Figure 5).

Indeed, in A-B+G+ mice with deleted alpha-syn, EEG coherence suppression was expressed to different extents in all inter-structural compositions (Figure 3). On average, relatively low but significant levels of EEG coherence were observed between both MC and Pt or MC and VTA (Figure 3A,B, respectively; two-way ANOVA: F_1,120_ = 8.1 and 10.2, *p* < 0.01 for both), while for the relations between the other brain areas of MC-SN, Pt-VTA, Pt-SN, and VTA-SN (Figure 3C–F, respectively), more evident deviations were characteristic (two-way ANOVA: F_1,120_ > 29.5, *p* < 0.001 for all; see Figure A3 for details).

In mice from the group lacking a full set of the synucleins (A-B-G-), the coherence suppression phenomenon disappeared in MC-Pt (Figure 4A; two-way ANOVA: F_1,96_ = 0.4, *p* > 0.6) and was expressed to a lesser extent in A-B+G- group (Figure 4B; two-way ANOVA: F_1,108_ < 6.7, *p* < 0.05 for MC-Pt, MC-VTA, and VTA-SN; F_1,108_ < 8.8, *p* < 0.01 for MC-SN and Pt-SN; see Figure A3 for details).

However, in mice without alpha- and beta-synucleins (Figure 4C), the disruption of the EEG coherence between the brain areas returned with maximal expression in the relations between MC or Pt and the dopaminergic areas, VTA, and SN (two-way ANOVA: F_1,126_ > 14.3, *p* < 0.001 for all, see Figure A3 for details).

The same was observed in both combinations with deleted beta-synuclein in mice from groups of A+B-G- (Figure 5A; two-way ANOVA: F_1,126_ > 11.4, *p* < 0.001 for all, with exception of VTA-SN relations: F_1,126_ = 6.0, *p* < 0.05) and A+B-G+ (Figure 5B; two-way ANOVA: F_1,114_ > 11.9, *p* < 0.001 for all, with exception of MC-Pt relations: F_1,114_ = 3.0, *p* > 0.0 f, see Figure A3 for details), and in the gamma-synuclein knock-out mice (Figure 5C; two-way ANOVA: F_1,90_ > 12.7, *p* < 0.001, for all, with exception of VTA-SN relations: F_1,90_ = 8.2, *p* < 0.01). All mentioned above conclusions are collected in Figure 6 and Figure A3, where the results of two-way ANOVA evaluation of differences between the full spectral profiles of the EEG coherence in KO mice and WT littermates are presented. Indeed, the F values for inter-structural coherences were distributed accordingly, such that the lowest F values were characteristic of both (i) MC-Pt (Figure 4a and Figure 5a) and VTA-SN (Figure 4f and Figure 5f) practically in all groups of KO mice (Figure 6, grey and blue bars, respectively), and (ii) at all inter-structural combinations in A-B+G- mice (Figure 4B). In contrast, the F values were evidently higher and dominated in the inter-structural relations with VTA or SN areas in KO mice from the other groups (Figure 3C–E, Figure 4C(b–e) and Figure 5(b–e)). Another specificity of these inter-structural relations was expressed in the domination of the coherence suppression effect in EEG at higher frequencies, in the alpha and beta bands (Figure 4C(b–e) and Figure 5(b–e), see Figure A2 for details). Finally, evidently enhanced suppression of coherent relations between MC and SN rather one between MC and VTA (Figure A4) directly reflects the lower susceptibility of VTA (versus SN) to disturbances in the synucleins’ family with exception of mice from A-B+G- group. This phenomenon was more selective in relations of Pt with VTA and SN and observed only in mice from A-B-G-, A-B-G+, and A+B-G- groups (Figure A5). 

## 4. Discussion

In the knock-out mice, we have shown that the loss of alpha-, beta-, and gamma-synucleins in all possible combinations was accompanied by disturbances in the brain “functional connectivity” that was expressed in significant attenuation of the coherence in EEGs of the cortex, putamen/striatum, and dopamine-containing regions: ventral tegmental area (VTA) and substantia nigra (SN). At the highest extent, this phenomenon was observed in the interconnections with VTA and SN and predominantly at the higher EEG frequencies, in the alpha and beta bands.

The delta and beta activities are well known to be associated in DA-accumulating brain areas characterized by “tonic” and “bursting” discharges, respectively, [24,25]. Thus, delta attenuation and beta enhancement in EEG recorded from VTA and SN expectedly correlate with tonic firing suppression and bursting pattern elevation in these brain areas. The coherent EEG relations in different brain structures, reflecting their “functional connectivity”, could be affected by modified compositions of the synucleins because of their associations with the changes in vulnerability and connectivity, thus shaping a propagation of the neurodegenerative disease [8]. The peculiarities in significant EEG coherence deviations in KO versus WT mice seem to be linked with specificities of both morphological inter-connections between analyzed brain structures and the EEG patterns generating by them. Indeed, well-known interconnections between SN and VTA [26] and similar compositions of “tonic” and “bursting” EEG patterns in these brain areas provide an explanation for relatively small differences in the coherent relations of their EEGs in mice from each of the groups (see Figure 3F, Figure 4f and Figure 5f). The same explanation seems to be adequate for minimal differences observed in the coherence between EEGs recorded from MC and Pt (see Figure 3A, Figure 4a and Figure 5a). Tremendous suppression in the beta bands of EEG coherence in inter-structural relations with DA-containing areas, VTA and SN, (Figure 4(b–e) and Figure 5(b–e), respectively) is seemingly associated with both a variety of distributions of the “fast” oscillations in these brain structures in KO mice from different groups (see [17]) and with a diversity in wiring architectures between the mesencephalic and forebrain regions [27]. Furthermore, some peculiarities in EEG coherence relations is expected to be associated with the lesser vulnerability of VTA versus SN that was demonstrated in experiments with Parkinson’s-inducing toxins [8] and confirmed in our study with selective deletion of different types of synucleins (see Figure A4 and Figure A5).

Surprisingly, across studied groups of KO mice, minimal changes in the EEG coherence between different brain structures versus WT littermates were mentioned in the group with combined absence of alpha- and gamma-syns (Figure 4B). This allow a preliminary conclusion that beta-syn on its own is able to support a native-like EEG coherence between the brain structures because of compensating increase of striatal dopamine [28,29], thus providing normal functioning of the brain. This is consistent the ability of beta-syn potentiate neurotransmitter uptake by synaptic vesicles in the absence of other synucleins [18]. Indeed, beta-syn mice from other groups with deleted either alpha-syn (Figure 3) or gamma-syn (Figure 4B) were characterized by evident suppression of the EEG coherence. Interestingly, this phenomenon was observed in beta-syn KO mice with concomitant deletion either alpha-syn (Figure 4C) or gamma-syn (Figure 5A). Thus, regardless of beta-syn is present or not, the suppression of EEG coherence is predominantly determined by the presence or the lack of one of the other synucleins. The specific role of the “flanking” (alpha and gamma) synucleins in their influences on the beta-syn depleted mice is readily visible in comparison of single- (A+B-G+) and triple-(A-B-G-) KO mice (c.f., Figure 3 and Figure 4A). Indeed, in the three Syn-KO mice, evidently diminished coherence suppression versus that in A+B-G+ mice was observed. This is an additional argument for that it is not the absence of any particular syn but an imbalance of syns causes tremendous changes in EEG coherence between the brain structures.

Neuronal bursting in the VTA/SN has been shown to produce significantly enhanced DA release in numerous targets in cortical and sub-cortical areas compared to an equivalent tonic activity [30]. Studies in both PD patients and animal models suggest that beta oscillations in the EEG can be affected by DA depletion [31], the typical characteristic of this neurodegenerative pathology. On the other hand, some adaptive modifications in dopaminergic synaptic mechanisms seem to be able to compensate a developmental ablation of dopaminergic neurons, thus protecting brain function [32]. One of the compensatory/adaptive mechanisms might be associated with DA receptor supersensitization [33]. Indeed, from our previous study of apomorphine (APO) effects in KO mice [17], significant differences in DA receptor sensitivity were indirectly demonstrated in mice with some peculiar combinations of the deleted synucleins. In particular, the most prominent differences versus WT mice were observed in double-syn KO (A-B+G-) and single-syn KO (A+B+G-) mice in all of studied cerebral structures and predominantly in alpha and beta 2 bands of the EEG frequency spectra (see “P” columns). Together, these and other APO-produced differences in KO and WT mice pointed to an enhanced sensitivity of DA receptors in KO mice. A peculiar role of each of the synucleins or their combinations in the revealed super-sensitivity of the DA receptors seems to be determined by their involvement in various stages of DA metabolism and its release, reuptake, and recycling [21,34,35]. Beta-syn has been shown to potentiate synaptic vesicle DA uptake, but in the absence of other synucleins [6], this seems to explain an extremely low level of EEG coherence suppression in A-B+G- mice (Figure 4B) in contrast to intensive expression of this phenomenon in other KO mice. Intensive coherence suppression was observed in double KO mice lacking alpha- and beta-syns (Figure 5C) with partial (by 20%) lowering of DA level [5] and in single-KO mice with deleted beta- or gamma-syns (Figure 3 or Figure 5B, respectively), with exception of the aforementioned A-B+G- mice (Figure 4B). Interestingly, that EEG coherence suppression was similarly expressed in single-KO mice with lacking either alpha- or gamma-syns (Figure 4C and Figure 5B, respectively) that is in line with evidence about similar involvement of these syns from VTA and SN in regulation of release and re-uptake in dopaminergic system [21]. The role of beta-syn as a stabilizing factor is readily visible in relatively enhanced EEG coherence suppression in the triple KO mice versus that in A-B+G- group (c.f. Figure 4A,B, respectively). Together, the results obtained in our study are in line with an idea that various phenomena revealed in mice losing the synucleins need to be analyzed in details for the understanding of complex effects of their mutual influences [36,37,38].

The main limitations in this study might be associated with a lack of evidence about both homogeneity of EEG sleep characteristics for the analyzed 30-min periods in mice from different groups and possible gender differences in the results obtained. We have proven stabilities of both EEG coherence between different cerebral structures in 30-min intervals (see “Section 3 Results”) and EEG values in all analyzed frequency bands and cerebral structures 60 min after saline injection (see in [17] figures 5–8, grey lines). Nevertheless, the computing of sleep patterns distribution in EEG by use of a “segmentation” approach allowing for their separation seems to be useful in further EEG experiments with KO mice, especially given evidence regarding sleep disturbances in mice with altered levels of synucleins (see e.g., [38]). Furthermore, gender-associated differences in DA neurons in VTA have been shown at the molecular, cellular, and behavioral levels [39] and in synucleinopathy models in mice [40]. This is in line with rising interest to the gender- associated diversity in the development of neurodegenerative pathologies. Finally, the results of our previous study about possible involvement of age-related adaptive mechanisms in intracerebral disturbances in mouse models of Alzheimer’s disease and amyotrophic lateral sclerosis [16] stimulate further analysis of this aspect in the synuclein knockout models.

## 5. Conclusions

We found that in KO mice versus WT littermates, the coherence (“functional connectivity”) between EEGs recorded from different brain structures was suppressed in practically all mice with various compositions of deleted syns: alpha, beta, and gamma. The coherence suppression was minimal in the inter-structural relations of MC-Pt and VTA-SN in all KO mice and in those lacking either all syns or only beta-syn. In other combinations of deleted syns, significant EEG coherence suppression in KO mice was dominant in relations with the dopamine containing areas, VTA and SN. Thus, deletions of syns were accompanied by significant attenuation of intra-cerebral EEG coherence depending on the imbalance of their different types. Further studies should unveil molecular and cellular mechanisms linking syns misbalance and changes in the EEG coherence in the brain, as well as if and how this approach can be applied for early differential diagnostics of synucleinopaties. 

## Figures and Tables

**Figure 1 biomedicines-12-00881-f001:**
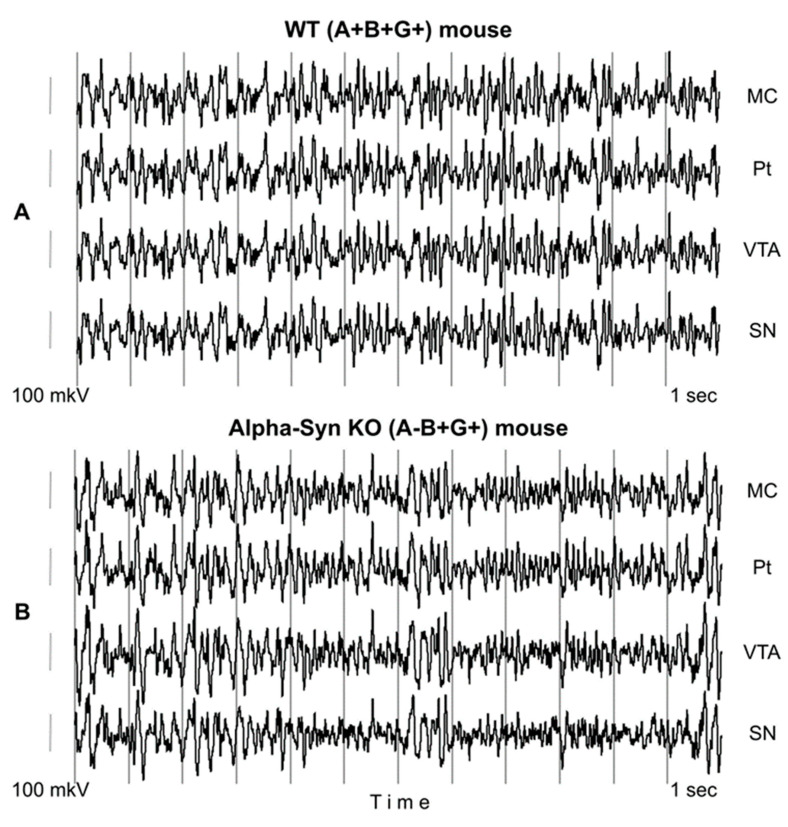
Typical 12-sec patterns of baseline EEG in WT (A+B+G+) mouse (**A**) and in alpha-syn KO (A-B+G+) littermates (**B**) recorded in different brain areas: MC, Pt, VTA, and SN. Time and amplitude calibrations are 1 sec and 100 µV, respectively.

**Figure 2 biomedicines-12-00881-f002:**
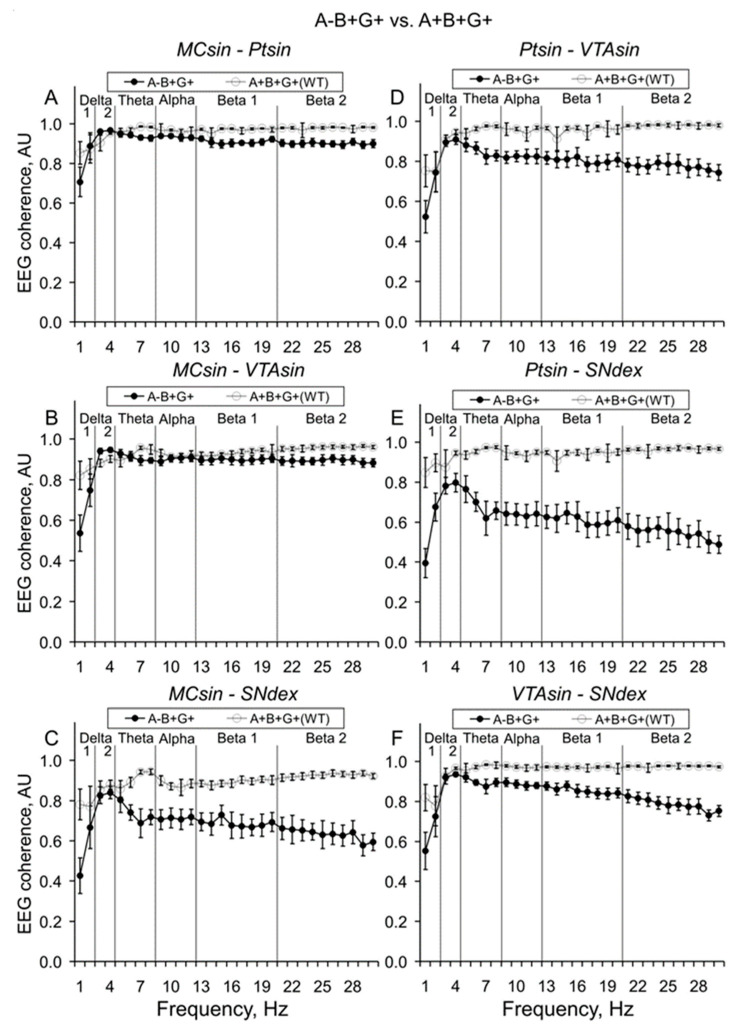
Averaged EEG coherences between different brain areas (MC. Pt, VTA, and SN) in 10-min baseline intervals in active 3-month-old wild-type mice (WT, grey lines) and alpha-syn KO littermates (A-B+G+, black lines). Inter-structural EEG coherences are specified on the plates of (**A**–**F**). Ordinate is the EEG coherence value in each of the 1 Hz bins analyzed in 0–30-Hz frequencies denoted on abscissa. Five vertical lines separate six “classical” EEG frequency bands, denoted on the plates.

**Figure 3 biomedicines-12-00881-f003:**
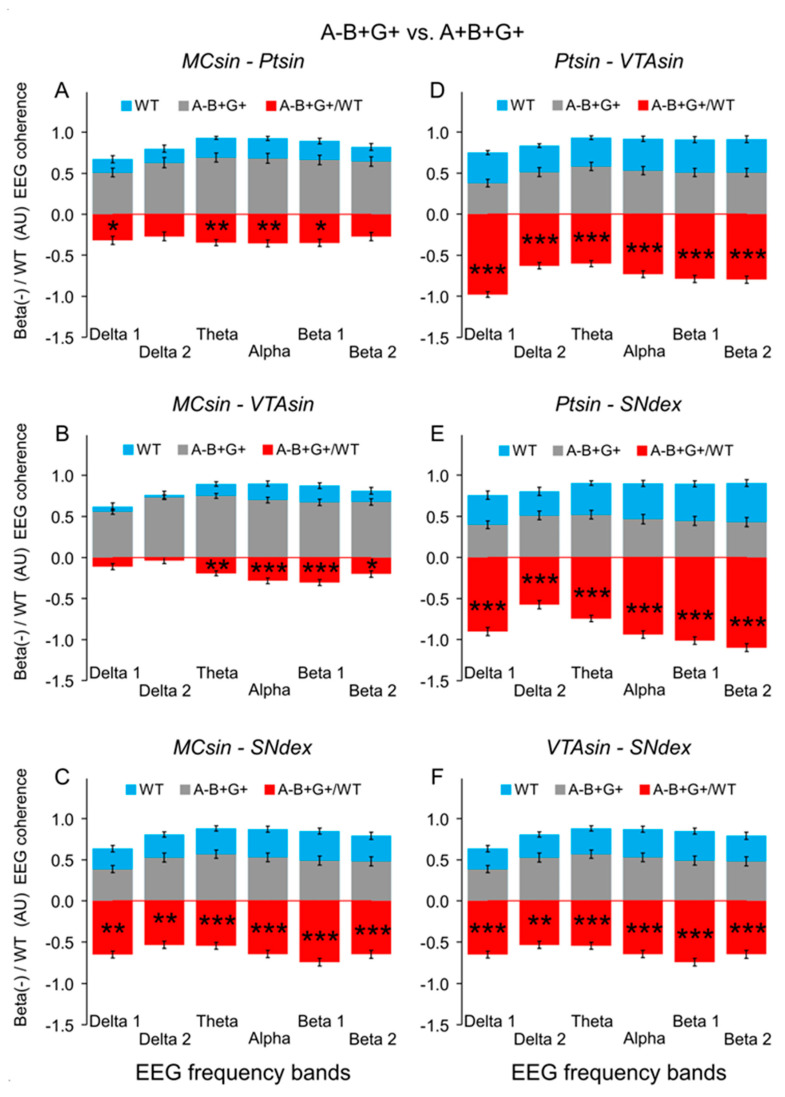
EEG coherences between MC, Pt, VTA, and SN averaged at 30-min baseline intervals (upward-directed ordinate) in WT mice (A+B+G+, blue bars) and in alpha-syn KO littermates (A-B+G+, gray bars). Red bars are relative differences between the above groups (downward directed ordinate) calculated as [(A-B+G+) − WT]/WT. Both ordinates are in arbitrary units. Different inter-structural relations specified on the plates of (**A**–**F**). Stars are significant two-way ANOVA differences between alpha-synuclein KO mice and their WT littermates (*, **, and *** are *p* < 0.05, <0.01, and 0.001, respectively) (see Figure A2 for details).

**Figure 4 biomedicines-12-00881-f004:**
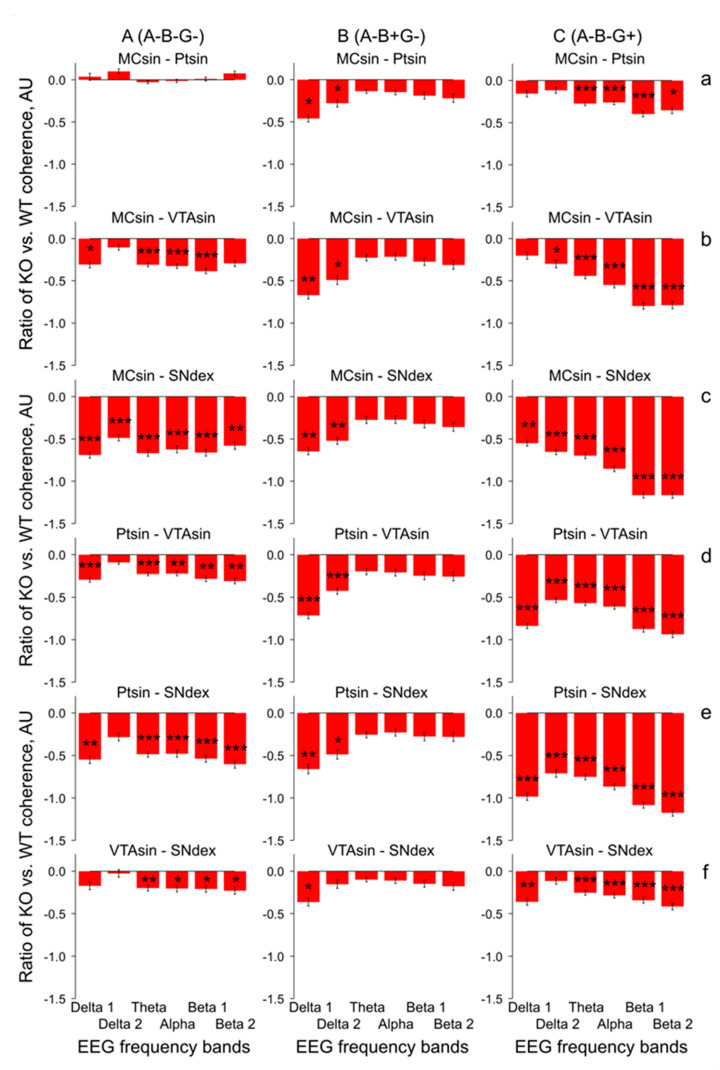
Relative differences between inter-structural coherences of baseline EEG recorded at 30-min intervals in synuclein knock-out mice (Syn-KO) with different combinations of deleted synucleins (alpha, beta, and gamma (**A**); alpha and gamma (**B**); alpha and beta (**C**)) and in WT littermates, calculated as [(Syn-KO] − WT)/WT, in arbitrary units. Different KO groups vs. WT littermates are specified on the plates of (**A**–**C**), while different inter-structural relations specified on the plates of (**a**–**f**). Stars are significant two-way ANOVA differences between Syn-KO mice and their WT littermates (*, **, and *** are *p* < 0.05, <0.01, and 0.001, respectively) (see Figure A2 for details).

**Figure 5 biomedicines-12-00881-f005:**
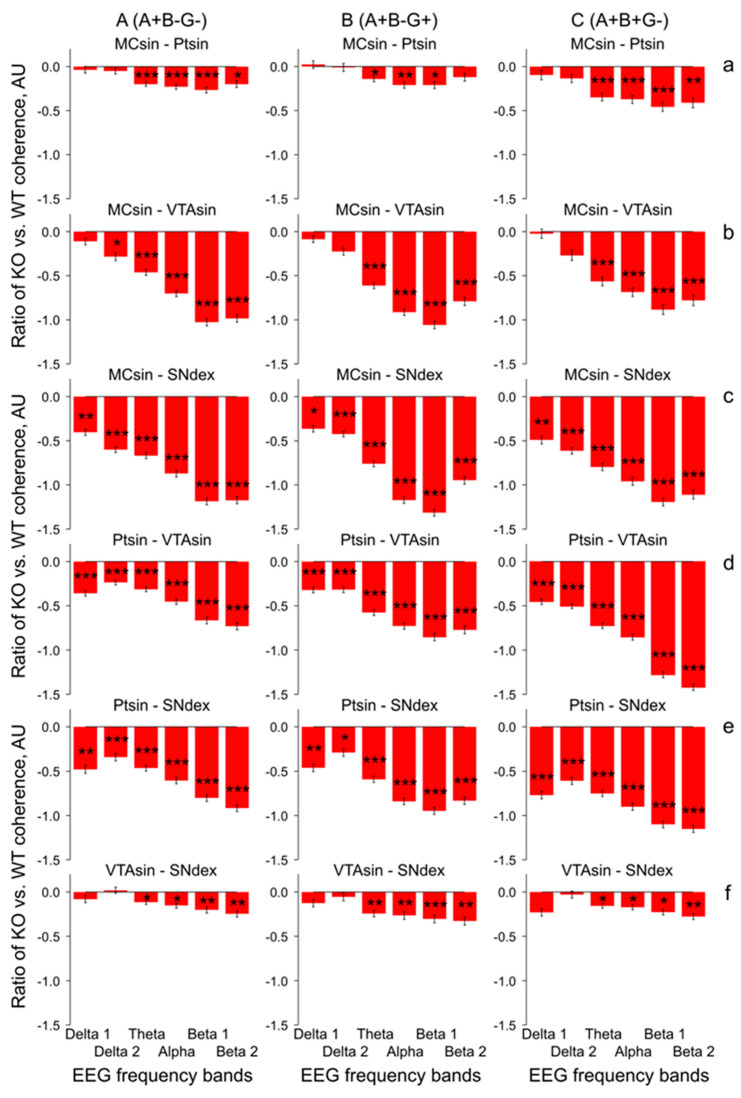
Relative differences between inter-structural coherences of baseline EEG recorded at 30-min intervals in synuclein knock-out mice (Syn-KO) with different combinations of deleted synucleins (beta and gamma (**A**); beta (**B**); gamma (**C**)) and in WT littermates, calculated as [(Syn-KO) − WT]/WT, in arbitrary units. Different KO groups vs. WT littermates are specified on the plates of (**A**–**C**), while different inter-structural relations specified on the plates of (**a**–**f**). Stars are significant two-way ANOVA differences between Syn-KO mice and their WT littermates (*, **, and *** are *p* < 0.05, <0.01, and 0.001, respectively) (see Figure A2 for details).

**Figure 6 biomedicines-12-00881-f006:**
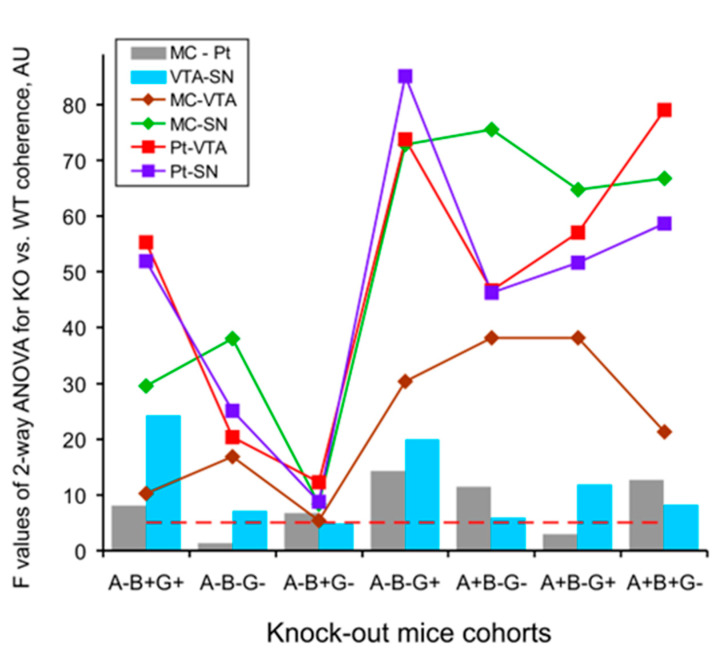
Distributions of the F values for the two-way ANOVA analysis of inter-structural coherence differences in full spectra of 30-min baseline EEG recordings from the brain areas of the 3-month-old KO mice with different combinations of deleted alpha-, beta-, and gamma-synucleins versus those in WT littermates. Horizontal red dashed line is a minimal significance (*p* < 0.05) threshold in this study (see Figure A3 for details).

## Data Availability

Data are contained within the current article.

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
