# Peer review of "Disruption of Electroencephalogram Coherence between Cortex/Striatum and Midbrain Dopaminergic Regions in the Knock-Out Mice with Combined Loss of Alpha, Beta, and Gamma Synucleins"

_biomedicines, 2024, doi:10.3390/biomedicines12040881_

Round 1

Reviewer 1 Report

Comments and Suggestions for Authors

In the current manuscript, baseline EEGs were recorded from wild-type and knock-out (KO) mice lacking alpha-, beta-, and gamma-synucleins. Further, EEG coherence was estimated between the cortex (MC), putamen (Pt), ventral tegmental area (VTA), and substantia nigra (SN). The authors concluded that deletions of the synucleins were sufficient to attenuate intra-cerebral EEG coherence, depending on the imbalance of different synucleins.

My concerns are as follows:

  1. Since male mice were used in the study, it's important to investigate or discuss whether female mice also exhibit similar alterations in coherence.
  2. As the study is based on prenatal deletion of synucleins, the authors should provide evidence for behavioral alterations accompanied by these deletions and further discuss the results.
  3. The author speculates on alterations in synaptic plasticity; however, immunohistological evidence for dopamine neurons or their terminals at MC and Pt will be necessary to validate conclusions.
  4. The results showed that intra-cerebral EEG coherence was altered in synuclein KO mice. It's important to discuss whether VTA and SN are equally susceptible to synucleinopathies and behavioral alterations.
  5. The major finding of the study is a significant difference in beta frequency. However, with only 30 minutes of baseline recording, it's highly unlikely that mice will enter all the sleep stages (NREM or REM sleep stages) where delta and theta frequencies are prominent. Therefore, to make the findings more conclusive, the authors should record for 2-4 hours for multiple trials select only trials with all three sleep stages, and then compare the frequency bands.
  6. Please also include the power vs. frequency comparison of the results to make the findings more authentic.
Comments on the Quality of English Language

I noticed a few grammatical errors in the text material. Please review it again.

Author Response

We are grateful to the reviewers for their constructive comments and we have addressed all their concerns in our revisions. Our responses are listed below

Reviewer 1

  1. Since male mice were used in the study, it's important to investigate or discuss whether female mice also exhibit similar alterations in coherence.

Author’s response: This aspect is briefly considered in the end of "Discussion"

  1. As the study is based on prenatal deletion of synucleins, the authors should provide evidence for behavioral alterations accompanied by these deletions and further discuss the results.

Author’s response: To reach this successfully, additional experiments with KO mice and WT littermates at different ages should be performed. This has being mentioned as a limitation in the end of "Discussion".

  1. The author speculates on alterations in synaptic plasticity….

Author’s response: Nowhere in the text, the term of “synaptic plasticity” was used. In Discussion, only well-known phenomenon of DA receptor hypersensitization, as a basis of compensatory (adaptive) mechanisms at attenuated ascending dopaminergic influence(s) from SN or VTA, has been considered.

  1. It's important to discuss whether VTA and SN are equally susceptible to synucleinopathies and behavioral alterations.

Author’s response: Different susceptibility of VTA and SN to disturbances in synucleins content is quite visible at the comparing of EEG coherences between MC-VTA and MC-SN or Pt-VTA and Pt-SN in each group of mice (see added Figures A3 and A4 and corresponding fragments in Results and Discussion).

  1. The major finding of the study is a significant difference in beta frequency. However, with only 30 minutes of baseline recording, it's highly unlikely that mice will enter all the sleep stages (NREM or REM sleep stages) where delta and theta frequencies are prominent. Therefore, to make the findings more conclusive, the authors should record for 2-4 hours for multiple trials select only trials with all three sleep stages, and then compare the frequency bands.

Author’s response: The relation between EEG recording duration and EEG coherence stability is briefly considered in the end of "Discussion". In addition, both WT and KO mice were characterized by mainly continuous motor activity that practically excluded sleep patterns or their combinations from EEG. Thus, the EEG coherence suppression in the beta frequency band in KO mice is unlikely associated with mechanisms of sleep and its disorders.

  1. Please also include the power vs. frequency comparison of the results to make the findings more authentic.

Author’s response: This aspect might be clarified comparing currently obtained data with those presented in our previous paper [13].

  1. I noticed a few grammatical errors in the text material.

Author’s response: Done

Reviewer 2 Report

Comments and Suggestions for Authors

The focus of the current study was brain synucleins influence on the pathology of Parkinson’s Disease (PD) in mice. Based on their prior research the purpose of the study to quantify using EEG coherence analyses the functional connectivity between brain regions. Accordingly, many of the method used were established in prior published studies including at least one in Biomedicines. The purpose of the study was accomplished by utilizing a total of 7 groups of knock-out (KO) mice that did not possess alpha-, beta- and gamma-synucleins and all combinations thereof. These groups were also compared to wild type (WT) mice. EEG coherence analyses were performed between motor cortex (MC), putamen (Pt), ventral tegmental area (VTA), and substantia nigra (SN) in all pairwise combinations. The numbers of groups and number of pairwise coherence comparisons made the findings somewhat complex, but the basic finding was that coherence was lower in KO mice compared with WT mice with the level of coherence suppression being more prominent is some of the combinations. 

Overall, the study seemed to be conducted carefully and had a strong design. The study was also relatively well-written with few grammatical or typographical errors. I think the study adds to the literature on the several interrelated topics associated with this research. I think the study will be of interest to readers of Biomedicine and researchers in several different fields. Another strength of the study not alluded to above was that the amount and presentation of the data was very good. I also like aspects of the coherence analysis such as the division of the frequency bands as there are different ways of doing that. I like the authors choice of bands. 

I don’t think the study has any fatal flaws. Therefore, most of my comments below are only moderate to minor concerns or questions of various types that the authors should address. 

  1. The sample size per group could potentially be viewed as somewhat low. Could the authors provide some reasoning of why the current sample size is adequate and how the number of mice per group was arrived at?
  2. Abstract line 18, I think the word “a” should be deleted and line 20 I think “an” should be “the”
  3. In general, it is believed that for frequency domain analysis and therefore coherence analysis that the EMG or in this case EEG signals should be stationary. Were mathematical procedures done to assure that the signals did not violate stationarity? If not is this issue important in this case to get valid coherence measures?
  4. In the coherence plots the dashed lines the authors show are the maximal and middle coherence values. Usually a dashed line on a plot like this is the statistical significance line (above chance or noise) for the coherence values. It may be best to include that type of information provide that information in the methods and in the figure legends (e.g. Figure 2). In other words, include the statistical significance line in for instance Figure 2. Relately, the authors do have the significance line on figure 6.
  5. Discussion Line 35 and perhaps elsewhere, I am not sure the term synchronous and variations of the word should be used or not. At least in some related fields (motor unit studies or EEG-EMG coherence studies) the term synchronization is a time domain measure (e.g. motor unit synchronization) and not a frequency domain measure. Since coherence is the cross spectrum of the autospectrum of the two signals and a frequency domain measure, perhaps the term synchronous should not be used and only the word coherence used?
  6. Bibliography: There are some inconsistencies in capitalization of the first letter of words that comprise the titles of some articles. Sometimes they are capitalized and sometimes not. For instance see the difference in references 9-13 where first letter is capital in the article titles versus references 3-8 as just some examples. Please proof and fix the bibliography.
  7. The authors should probably add a separate Limitations section near the end of the Discussion.

Comments on the Quality of English Language

minor proofreading and a few format inconsistencies in bibliography

Author Response

We are grateful to the reviewer for constructive comments and we have addressed all his/her concerns in our revisions. Our responses are listed below

Reviewer 2

  1. Could the authors provide some reasoning of why the current sample size is adequate?

Author’s response: This aspect of statistics is undeservedly ignored in the literature (for review see, e.g., Int J Psychophysiol. 2017; 111: 33-41). In our study, quite realistic for EEG effect sizes (0.75 - 0.9) and power (0.8) were chosen with calculated by G*Power (version 3.1.9.4) reasonable sample sizes. This has been added in the main text.

Table 1

F tests - ANOVA: Repeated measures, between factors

Analysis:

A priori: Compute required sample size

Input:

Effect size f(V)

0.75

0.90

1.20

α err prob

0.05

0.05

0.05

Power (1-β err prob)

0.80

0.80

0.80

Number of groups

2

2

2

Number of measurements

18

18

18

Output:

Noncentrality parameter λ

10.7

12.3

16.4

Critical F

5.32

5.99

7.70

Numerator df

1

1

1

Denominator df

8

6

4

Total sample size

10

8

6

Actual power

0.815

0.830

0.851

  1. Abstract line 18, I think the word “a” should be deleted and line 20 I think “an” should be “the”

Author’s response: Done

  1. Were mathematical procedures done to assure that the signals did not violate stationarity?

Author’s response: Before EEG coherence evaluation, the EEG recordings were usually analyzed by use of relatively simple and efficient KPSS test to determine time duration of the epochs characterized by stationary EEG. Within them, optimal duration of 12 sec was chosen for subsequent evaluation of EEG coherence that coincided with characteristic duration revealed in our previous "coherence" studies. This has been added in the main text.

  1. In the coherence plots the dashed lines ….

 Author’s response: In Figures 2-5, the horizontal dashed lines have been removed, for clarity.

  1. ….perhaps the term synchronous should not be used and only the word coherence used?

Author’s response: Done

  1. There are some inconsistencies in capitalization of the first letter of words that comprise the titles of some articles.

Author’s response: Done

  1. The authors should probably add a separate Limitations section near the end of the Discussion.

Author’s response: Done

  1. English Language minor proofreading and a few format inconsistencies in bibliography

Author’s response: Done

Reviewer 3 Report

Comments and Suggestions for Authors

This is an interesting article, but there are some points that need revision:

The literature review is poor. Please add relevant references from relevant research.

The hypotheses are not clearly presented and supported by relevant literature.

The figures are difficult to follow. Please explain all figures in the text.

Why was Bonferroni's used as a posthoc test? Why not other alternatives?

Comments on the Quality of English Language

Moderate English language is needed throughout the text.

Author Response

We are grateful to the reviewer for constructive comments and we have addressed all his/her concerns in our revisions. Our responses are listed below

Reviewer 3

The literature review is poor. Please add relevant references from relevant research.

Author’s response: Done, at first approximation

The hypotheses are not clearly presented and supported by relevant literature.

Author’s response: Done, at first approximation

The figures are difficult to follow. Please explain all figures in the text.

Author’s response: Done

Why was Bonferroni's used as a posthoc test? Why not other alternatives?

Author’s response: We prefer the Bonferroni correction because of its efficacy as a post-hoc procedure following ANOVA when it applies to a small number of compared means (Armstrong RA. When to use the Bonferroni correction. Ophthalmic Physiol Opt 2014; 34: 502–508. doi: 10.1111/opo.12131).

Moderate English language is needed throughout the text.

Author’s response: Done

Round 2

Reviewer 1 Report

Comments and Suggestions for Authors

Thank you for appropriately addressing my concerns.